# Wind Farm Layout Optimization with Different Hub Heights in Manjil Wind Farm Using Particle Swarm Optimization

Menova Yeghikian [1], Abolfazl Ahmadi [1,*], Reza Dashti [1], Farbod Esmaeilion [2], Alireza Mahmoudan [3], Siamak Hoseinzadeh [4,*] and Davide Astiaso Garcia [4]

1 Department of Energy Systems Engineering, School of New Technologies, Iran University of Science and Technology, Tehran 13114-16846, Iran; ahmadi.abolfazl@gmail.com (M.Y.); rdashti@iust.ac.ir (R.D.)
2 Department of Mechanical Engineering, K.N. Toosi University of Technology, Tehran 19967-15433, Iran; farbodesmailion@gmail.com
3 Department of Aerospace Engineering, K.N. Toosi University of Technology, Tehran 16765-3381, Iran; aremahmoudan@gmail.com
4 Department of Planning, Design, Technology of Architecture, Sapienza University of Rome, Via Flaminia 72, 00196 Rome, Italy; davide.astiasogarcia@uniroma1.it
* Correspondence: a_ahmadi@iust.ac.ir (A.A.); siamak.hoseinzadeh@uniroma1.it (S.H.)

**Abstract:** Nowadays, optimizing wind farm configurations is one of the biggest concerns for energy communities. The ongoing investigations have so far helped increasing power generation and reducing corresponding costs. The primary objective of this study is to optimize a wind farm layout in Manjil, Iran. The optimization procedure aims to find the optimal arrangement of this wind farm and the best values for the hubs of its wind turbines. By considering wind regimes and geographic data of the considered area, and using the Jensen's method, the wind turbine wake effect of the proposed configuration is simulated. The objective function in the optimization problem is set in such a way to find the optimal arrangement of the wind turbines as well as electricity generation costs, based on the Mossetti cost function, by implementing the particle swarm optimization (PSO) algorithm. The results reveal that optimizing the given wind farm leads to a 10.75% increase in power generation capacity and a 9.42% reduction in its corresponding cost.

**Keywords:** wind farm; optimization; particle swarm optimization; wind farm layout optimization

## 1. Introduction

Concerns over climate change as well as the dwindling resources of fossil fuels have made experts replace renewable energy sources more than before [1–5]. Therefore, the recent years have seen a rapid climb in the number of renewable energy-based power plants. Among all types of renewable energy resources, wind energy plays a crucial role in technical and economic approaches. Over the recent years, wind energy has become a promising alternative renewable resource for fossil fuels [6–10]. In this regard, wind farms or wind power plants have been developed all over the world to utilize this cost-effective energy source even in remote areas with high potentials [11–13]. The applications of wind energy are growing rapidly. However, there are still some challenges facing these applications. The main issue associated with wind energy is that the actual power generated by wind farms is less than its theoretical power capacity due to the wake effect, wind velocity variations, angle variations, wind turbine inefficiencies, and transmission line problems [14–16]. The wake effect can be characterized as wind speed reduction and turbulent flow creation downstream of wind turbines [17–20]. As a result, this impact can reduce power generation by 10–20%. An enhanced layout of wind turbines can significantly improve power output and reduce its associated charges [14,21]. The power generated in a wind turbine is related to the perfect square of the receiving wind speed. Thus, the wind speed reaching each turbine is preferred to the extent its maximum. Nevertheless,

the wind speed reaching downstream turbines is reduced in magnitude, causing a sharp drop in power generation of those turbines (as they are being under the wake effect of upstream turbines). Therefore, the layout of wind turbines should be arranged in a way to minimize the impact of the wake effect. This procedure is known as wind farm layout optimization (WFLO) [22–26]. By doing so, the generated power from each turbine can reach a maximum value with lower associated costs.

The first priority in constructing a wind farm is to find a proper location [25,27–29]. This is due to the scarcity of lands and shortage of capital [30–32]. Consequently, wind turbine positioning comes next. As a result, to harness the highest possible power from a certain power plant, a comprehensive assessment on the placement of wind turbines should be taken into account. Wind farm layout optimization is a crucial subject in wind energy literature [33–35]. At this point, numerous researchers have been concentrated on the WFLO problem [16,36–38].

Patel [39] stated that in designing a wind farm layout, appropriate distances between wind turbines should be estimated. According to Patel's study, the optimal arrangement is in rows of 8–12 rotor diameters separately along the wind direction and 1.5–3 rotor diameters apart along the crosswind direction. One of the first research attempts for optimizing a wind farm layout was made by Mossetti et al. [40]. By using the genetic algorithm, they minimized the objective function value, which was the unit cost per power production. They implemented a simple, empirical cost model. In 2013, Samorani et al. [41] underlined the importance of the optimization of wind farm configurations involving optimally placement of wind turbines to diminish the wake effect. This problem has drawn the attention of the scientific community. However, existing approaches are not fully responding to the needs of wind farm developers, mainly since they do not usually address the challenges associated with construction and logistics. Eroğlu et al. [42] used a particle filtering approach to achieve an optimal layout of a specific wind farm. The boundary of the wind farm and distances between turbines were regarded as two main constraints. The results indicated that the particle filtering approach can compete with the ant colony and evolutionary strategy algorithms. Chen et al. [43] investigated the effect of using wind turbines with different hub heights on the overall output power. The nested genetic algorithm was used to analyze three different wind conditions. The results of this study demonstrated that by using different hub heights of wind turbines, power generation increased (compared with a wind farm having the same number of turbines). Shakoor et al. [44] proposed a novel method called definite point selection (DPS) that could find the optimum placement of turbines. The DPS method was approved to be more effective than the earlier proposed methods. Gao et al. [22] presented a 2D analytical wake effect model based on the Jensen's wake model and Gaussian function. In this study, wind farm efficiency dropped to 77.83% from 96.83% for a collection of 38 wind turbines within a large wind farm. Wang et al. [45] addressed more complex wind farm boundaries by which a new constraint handling method was introduced. This study stated that the unrestricted coordinate method, under the sequential land plot scenario, generates optimal outcomes, with the lowest energy cost and highest efficiency. Parada et al. [46] used the Gaussian wake model to calculate wind speed loss. In this study, the cost of energy was optimized by the genetic algorithm. They asserted that the use of a more robust wake model in the WFLO problem did not lead to greater efficiency in real wind farm cases.

Sun et al. [47] adopted a conceptual 2D wake model to calculate wind losses caused by the wake effect. In this study, the cost of energy (COE) was used as a criterion to compare the effectiveness of this novel method. The results indicated that the optimization method used in this study can reduce the COE down to 1.02 HK\$/kWh.

Vasel Behagh et al. [48] studied the effect of height optimization of turbines on the annual energy production (AEP). In this study, they compared two wind farms with an equal number of turbines (also similar types and positions). In one wind farm, all turbine heights were identical, while the other one had alternating rows of tall and short wind turbines. The results exhibited that the vertically staggered configuration generated more

power output by 5.4%. Hou et al. [49] proposed an optimization technique for offshore wind farms. In this study, they attempted to find the optimized layout of turbines in order to achieve maximized power generation. The PSO algorithm with multiple adaptive methods (PSO-MAM) was used as an optimization algorithm. The results of this study indicated that the aforementioned method can suggest a layout that increases the power output by 3.84%. Tian et al. [50] investigated the optimal tip speed ratio and pitch angle for wind turbines by an exhaustive search (a brute-force search). A solution for this problem was introduced considering the estimation error of the wake model, which was to yield the optimal control curves for each wind turbine. In that study, the annual energy production was increased by 1.03%. Mir Hassani et al. [51] studied the effect of using different hub heights on the total power generation of a wind farm. They presented a mathematical model for wake effects considering wind turbines with different hub heights as well as a new optimization model. Abdelsalam et al. [52] aimed to optimize a wind farm layout by the binary real coded genetic algorithm (BRCGA) based on a local search (LS), gathering robust single wake models with suitable wake interaction modeling. The model used in this study was the Jensen wake model alongside the sum of squares model.

Kirchner-Bossi et al. [53] introduced a new technique for wind farm layout problems using the Gaussian wake model. This methodology was applied to two real wind farms and was also compared with the Jensen model. Stanley et al. [54] optimized some wind farms using a coupled optimization method. The coupled optimization led to a reduction in the cost of energy by 2–5% compared to sequentially optimization of wind farms with turbine spacing of 8.5–11 rotor diameters. Some of the wind farms in that study also exhibited an additional 10% reduction of energy costs.

Pratt et al. [55] compared several wake effect models using analytical and CFD methods for a wind farm at Block Island. The results showed that a value in the higher range of the examined WDC (0.06 and 0.07) and TI (12% and 14%) values represented a better comparison to the observed data. Diaz et al. [56] developed the actuator disc (AD) model which was the most common simplified wind turbine model, based on the Open FOAM open-source software. Results demonstrated that values for low and high wake impact situations were improved with 2.5% and 1.3%, respectively. Patel et al. [57] introduced a novel method called the geometrical pattern-inspired placement methodology to find the layout of turbines with maximum total power output at Kutch-India. The enhanced passing vehicle search (PVS) algorithm was used in this study. The results showed that the power output was improved by 4.29%.

The purpose of this original research is to optimize the 3D layout of turbines at the Valfajr Wind site in Manjil, Iran. In other words, this is an attempt to identify the best placement of turbines and their heights as the power plant reaches its optimal operating level [58]. Initially, by considering the wind and geographic information of the region, the wake effect is analyzed with the Jensen method [59,60]. Next, the objective function, which is the cost of total power generation [40,61,62], will be estimated. Finally, the objective function will be optimized using the particle swarm optimization algorithm.

## 2. Materials and Methods

In this study, the Jensen's method is used to model the wake effect. As shown in Figure 1, when wind passes through a rotor blade, its speed falls and the waking zone spreads from the wind flow, similar to a cone [63–65]. The radius of this cone can be calculated.

As shown in Figure 1, if the wind turbine zone is located downstream of the wake cone, the wake and the downstream turbine will overlap. In case the region is generated by an upstream wind turbine, the wind speed will experience reduction [6,66]. Additionally, Figure 2 presents the wake shadow area.

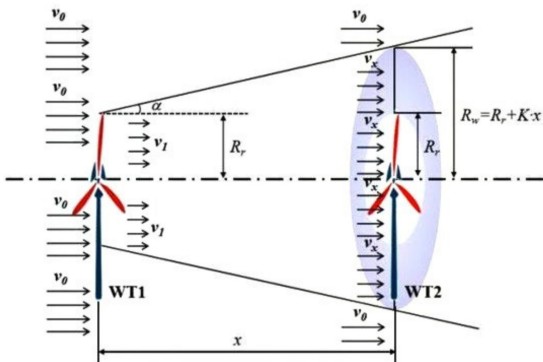

**Figure 1.** The wake effect produced by a downstream unit of shadow caused by the existence of an upstream wind turbine unit.

$$R_w = R_r + kx \tag{1}$$

In the above equation, $R$ is rotor's radius, $x$. is the distance of turbines, and $k$ is the expansion rate that takes a value between 0.04 to 0.08 which is calculated by [67,68]:

$$k = \frac{0.5}{\log\left(\frac{z}{z_0}\right)} \tag{2}$$

where $z$ is the hub height and $z_0$ is the roughness length. Wind flow velocity of the wake is computed as [69,70]:

$$V_{single} = V_0\left(1 - \left(1 - \sqrt{1 - C_T}\right)\left(\frac{R_r}{R_w}\right)^2\left(\frac{A_{shad}(ud)}{A_0}\right)\left(\frac{V_z}{V_{ref}}\right)\right), \quad k = \frac{0.5}{\log\left(\frac{z}{z_0}\right)} \tag{3}$$

where $A_0$ is the area created by the rotor blade system and $A_{shad}(ud)$ refers to an area covered by the downstream turbine and the shaded area of the upstream turbine, which is calculated by Equation (4) [71]:

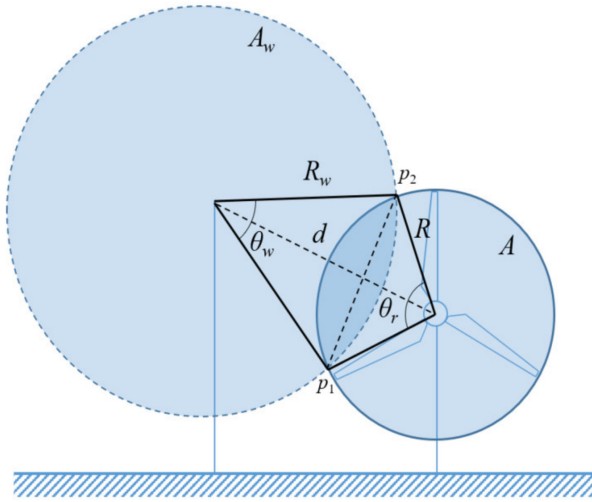

**Figure 2.** Wake shadow area [71].

$$A_{shad}(ud) = \begin{cases} \pi R^2 & d < R_w - R \\ \frac{1}{2}R^2(\theta_r - \sin\theta_r) + \frac{1}{2}R_w^2(\theta_w - \sin\theta_w) & R_w - R < d < R_w + R \\ 0 & d > R_w + R \end{cases}$$

$$\theta_r = 2\cos^{-1}\frac{R^2 + d^2 - R_w{}^2}{2Rd_w} \qquad \theta_w = 2\cos^{-1}\frac{R_w{}^2 + d^2 - R^2}{2R_w d} \tag{4}$$

For calculating wind speed originated from multiple upstream turbines reaching a downstream wind turbine, the following equation is used (Equation (5)):

$$V_d = V_{in}\left(1 - \sum\left(1 - \sqrt{1 - C_T}\right)\left(\frac{R}{R_w}\right)^2\left(\frac{A_{shad}(ud)}{A_0}\right)\left(\frac{V_z}{V_{ref}}\right)\right) \tag{5}$$

Shaded wind turbines and the number of shaded units are different and they both depend on the wind direction and the geometrical layout of wind turbines. It is illustrated in the following figure for two different directions of wind. As shown in Figure 3, the area covered by the vortex cone changes with wind direction [51].

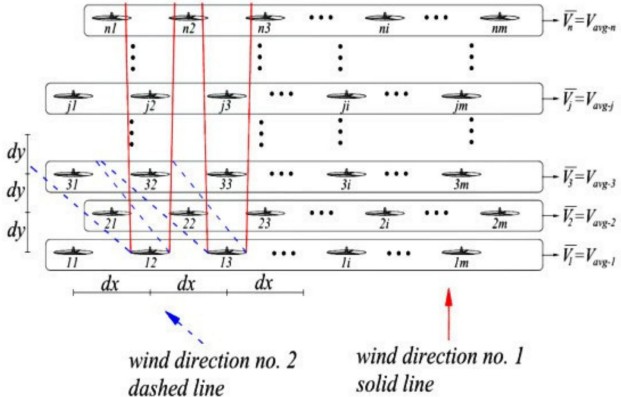

**Figure 3.** The effect of wind direction on the wake effect by upstream wind turbines for two different wind directions [51].

The wake factor can be defined as [37,72]:

$$C_w = \frac{Total\ power\ of\ wind\ farm\ considering\ wake\ effect}{Total\ power\ of\ wind\ farm\ without\ wake\ effect} \tag{6}$$

which can be formulated as:

$$C_w = \frac{\sum_{j=1}^{n}\sum_{i=1}^{m}P\left(V_{ij}\right)}{mn \times P_n} \tag{7}$$

where $m$ is the number of turbines in a row, $n$ is the number of rows, and $V_{ij}$ is the wind speed fraction perpendicular to the $i$-th turbine in the $j$-th row. In addition, $P_n$ stands for the power generated by a turbine with wind speed of $V_{ij}$. When the wake factor is equal to one, it means there is no wake effect.

The following relation is used to obtain wind speed at any hub height [66,73]:

$$V_z = V_{ref}\frac{\log\frac{z}{z_0}}{\log\frac{z_{ref}}{z_0}} \tag{8}$$

where $V_{ref}$ represents the reference wind speed and $z_0$ is the roughness length.

The cost function considered in this study is the same as Mossetti et al. [40]. Through this function, the annual plant's total cost by using the number of turbines ($N_t$) can be calculated with decent accuracy.

$$Cost_{tot} = N_t\left(\frac{2}{3} + \frac{1}{3}e^{-0.00174N_t^2}\right) \tag{9}$$

The objective function for the optimization problem in this study is defined as:

$$Objective_{Function} = \frac{Cost_{tot}}{P_{tot}} \quad (10)$$

where $P_{tot}$ stands for the total power generation in the wind farm and can be calculated as:

$$P_{tot} = \sum_{i=1}^{N_t} P_i \quad (11)$$

where:

$$P_i = \frac{1}{2}\rho\pi R^2 V_{in}^3 C_p \quad (12)$$

where $\rho$ is air density, $R$ is the rotor radius, $V_{in}$ is wind speed, and $C_p$ is the power coefficient of the wind turbine.

## 3. Particle Swarm Optimization

In this research, particle swarm optimization is used to optimize the objective function. Due to the appropriate and adapted nature of the algorithm to achieve an optimal solution, the algorithm has fast speed [4,66]. Figure 4 indicates the motion pattern in the particle swarm algorithm.

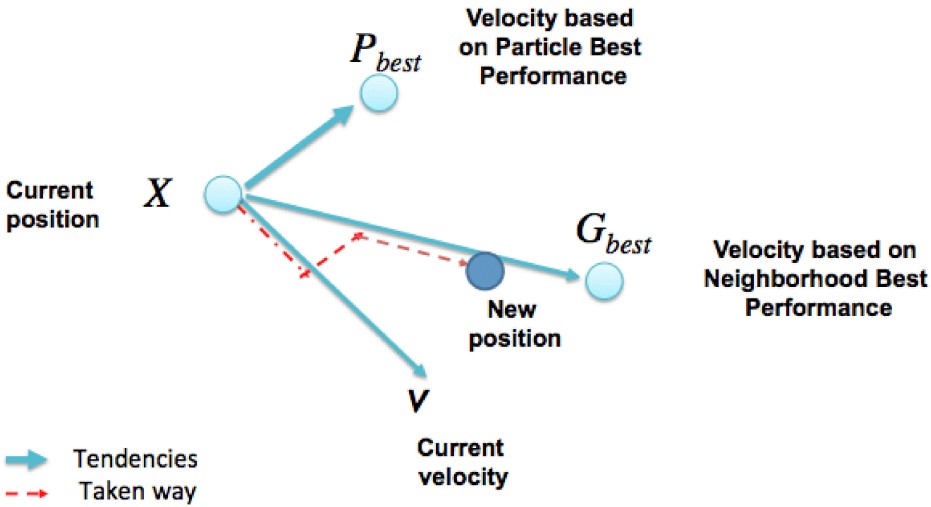

**Figure 4.** The motion pattern in the particle swarm algorithm [74].

In each step, the velocity of particles becomes updated from the following relationships:

$$V_i(t) = w \times V_i(t-1) + c_1 \times rand_1 \times (P_{i.best} - X_i(t-1)) + c_2 \times rand_2 \times \left(P_{g.best} - X_i(t-1)\right) \quad (13)$$

$$X_i = X_i(t-1) + V_i(t) \quad (14)$$

In Equation (13), $w$ is the inertia weight factor which indicates the effect of the iteration speed vector $V_i(t)$ on the velocity vector in the current iteration $V_i(t+1)$, $c_1$ is the constant learning coefficient moving in the direction of the best personal particle value, $c_2$ is the constant learning coefficient moving along the path of the best particle found among the entire population, and $rand_1$ with $rand_2$ are two random numbers with uniform distribution from 0 to 1.

*Steps of the Implementation of the Particle Swarm Algorithm*

(1) Random production of particles: the stochastic production of the initial population is simply the random assignment of particles with uniform distribution in the solution space.

(2) Purpose of the objective function: At this stage, each particle representing a problem-solving must be evaluated. Depending on the provided equation, the evaluation method will be different.

(3) Recording the best location for each particle and the best position among the whole article: at this step, the amount of target function obtained for all particles is compared with the best amount of cost obtained for each particle.

(4) Updating the speed vector of all particles: after calculating the best particle for each this method, the velocity vector available for each particle is updated by the best position of each particle, the current position, and the best position among all particles.

(5) Convergence test: There are various methods for investigating the algorithm. For example, a certain number of iterations can be found from the beginning. Another method often used for convergence test of the algorithm is that if there is no change in the value of the best particles in a sequence of consecutive iterations, then the algorithm ends.

## 4. Manjil Wind Farm

In this study, the Valfajr wind site at Manjil wind farm is investigated. The site was built on a site with an approximate size of $2 \times 2$ km$^2$. There are 21 Nordtank wind turbines at the site. Of those, one has 500 kW, 5 have 550 kW, and the rest has 300 kW power generation capacity [75]. Wind rose shows prevailing wind direction from north to south. In most cases (27%), the wind blows from the northern areas of Alborz Mountain to the southern regions. Furthermore, it does not speed up in 40% of the time. In the city of Manjil, the maximum wind speed experienced in the year is 18 m/s. Based on Ref. [75], the overall average wind speed in the year is 15 m/s. The associated wind rose is illustrated in Figure 5. In addition, Table 1 presents the detailed specifications of the wind turbines.

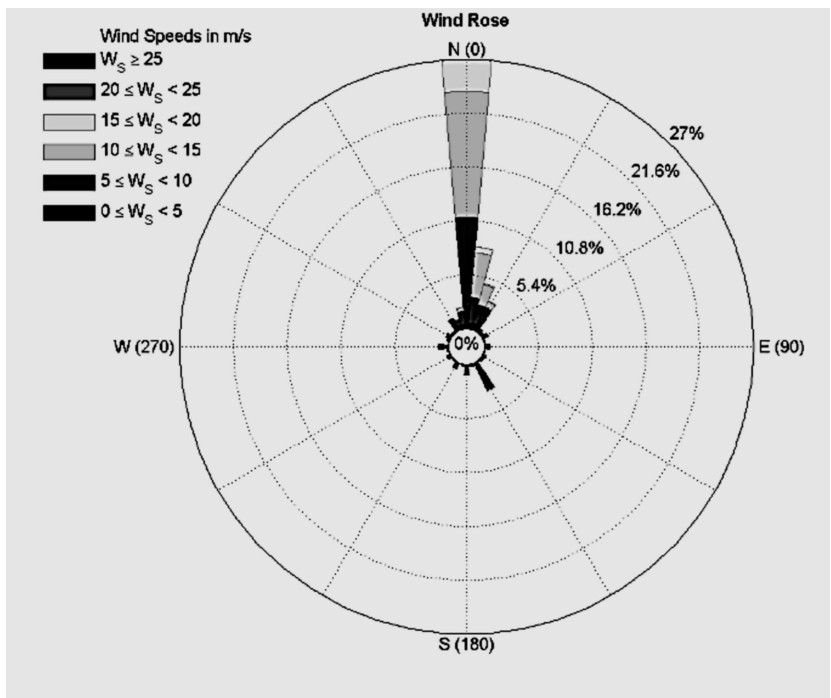

**Figure 5.** Wind Rose [76].

**Table 1.** Detailed specifications of the wind turbines.

| Turbine Specifications | |
|---|---|
| Rotor radius | 50 m |
| Reference temperature | 293 K |
| Reference air density | 1.225 kg/m$^2$ |
| Roughness length | 0.075 m |
| Thrust coefficient | 0.025 |
| Power coefficient | 0.45 |

In this research, the GID software was first used; the grid required for the plant and the placement of turbines were specified. Then, by using MATLAB, the required values for computing input values were set, and the initial population was generated to optimize the particle swarm optimization algorithm. The associated flowchart of PSO can be found in Figure 6. Next, making sure that the criteria point subsequent populations were produced. Finally, after accomplishing the criteria point, the results were achieved.

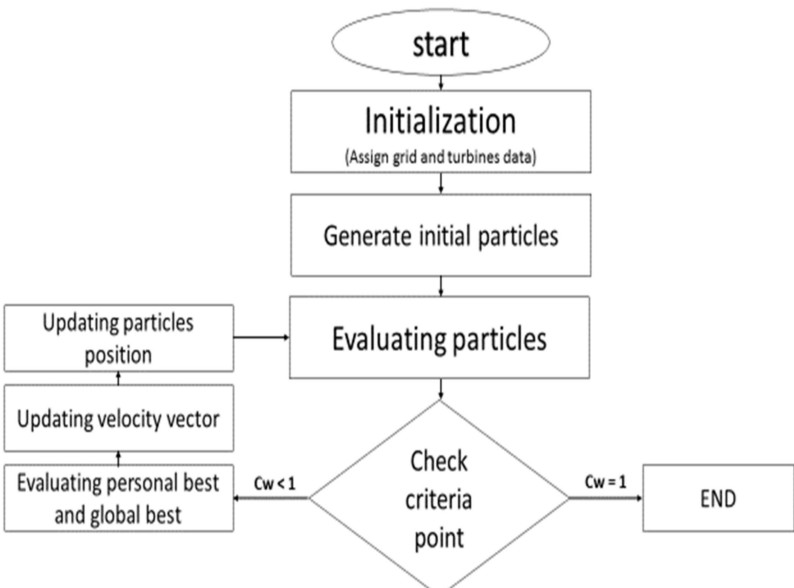

**Figure 6.** Particle swarm optimization (PSO) Flowchart.

## 5. Validation

The detailed specifications of wind turbines from the literature are presented in Table 2. To verify the accuracy of the algorithm used in this study, the obtained results are compared with the literature [43]. In that study, they analyzed a plant with a dimension of $1000 \times 1000$ m$^2$ and wind turbines with 21 kW power generation capacity.

**Table 2.** The detailed specifications of wind turbines from the literature [43].

| Turbine Specifications | |
|---|---|
| Roughness length | 0.3 m |
| Rotor radius | 40 m |
| Power coefficient | 0.4 |
| Thrust coefficient | 0.8888 |
| Wind speed | 13 m/s |

In the first scenario of this study, the speed and direction of wind are considered constant. The results are listed in Table 3. As can be seen, there are less than 5% difference between the two results.

**Table 3.** The validation of the objective function for cost per power in the first scenario.

| Number of Turbines | Song Function (Cost/Power) | Present Research's Function | Error (%) |
|---|---|---|---|
| 10 | 1.005687 | 1.045914 | 4 |
| 15 | 1.011899 | 1.06047 | 4.8 |
| 20 | 1.033428 | 1.037166 | 2.3 |
| 25 | 1.09575 | 1.136856 | 3.75 |

In the second scenario, 36 wind directions with different wind speeds and probability are considered. The wind directions are uniformly distributed with intervals of 10 degrees and the wind speed of 12 m/s. The results are shown in Table 4. From the error column, it is plain to see that the results obtained in this study are in a good agreement with that of [43].

**Table 4.** The validation of the objective function for cost per power in the second scenario.

| Number of Turbines | Song Function (Cost/Power) | The Function of This Research | Error (%) |
|---|---|---|---|
| 10 | 1.234943 | 1.25964 | 2 |
| 15 | 1.308199 | 1.6736 | 5.1 |
| 20 | 1.397825 | 1.480278 | 5.7 |
| 25 | 1.504184 | 1.562847 | 4 |

## 6. Results and Discussion

In this research, three scenarios with different number of turbines (i.e., 10, 15, and 21) are considered. Wind speed is assumed to be 15 m/s, and the value each parameter of PSO takes is shown in Table 5. The size of the plant is $2 \times 2$ km$^2$ for all scenarios. Due to the minimum rate of turbines at a length of 4 times the radius of rotors, equivalent to 100 m, the grid is divided into 100 points. Furthermore, possible heights of turbines are 40, 50, and 60 m. The criteria point is where the wake factor becomes 100%.

**Table 5.** The PSO specification for the optimization procedure.

| PSO Specifications | |
|---|---|
| C1 | 1 |
| C2 | 1 |
| W | 0.05 |
| Number in the population | 5 |

### 6.1. First Case

In this case, there are 10 wind turbines, with the characteristics stated before. The placements and the corresponding height of turbines are shown in Figures 7 and 8, respectively. In this case, Table 6 provides the coordination of each turbine.

**Table 6.** Turbine coordinates in Case 1.

| Turbines Coordinates in Case 1 | | | | | | | |
|---|---|---|---|---|---|---|---|
| Number of Turbine | X (m) | Y (m) | Z (m) | Number of Turbine | X (m) | Y (m) | Z (m) |
| 1 | 350 | 0 | 40 | 6 | 0 | 1000 | 40 |
| 2 | 900 | 200 | 50 | 7 | 300 | 850 | 50 |
| 3 | 150 | 250 | 50 | 8 | 600 | 950 | 40 |
| 4 | 450 | 400 | 40 | 9 | 600 | 400 | 50 |
| 5 | 850 | 700 | 50 | 10 | 750 | 700 | 40 |

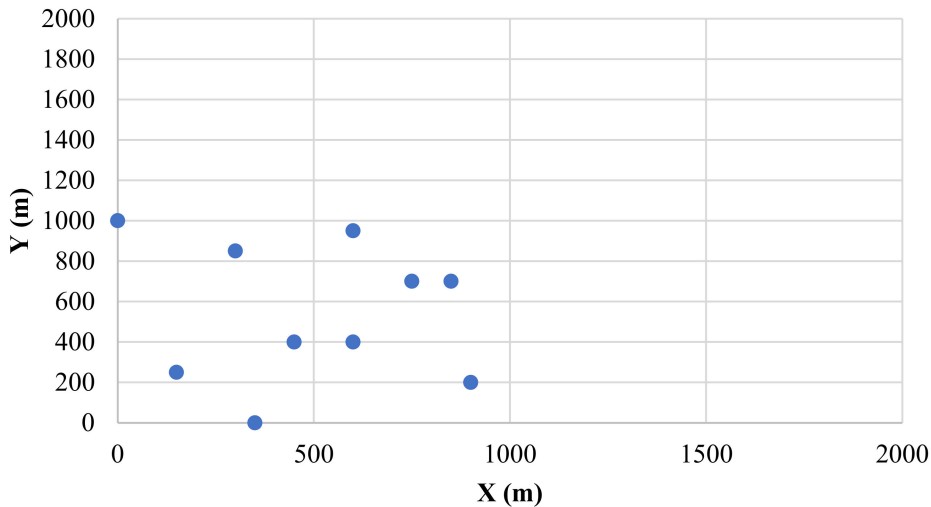

**Figure 7.** Turbine placements in Case 1.

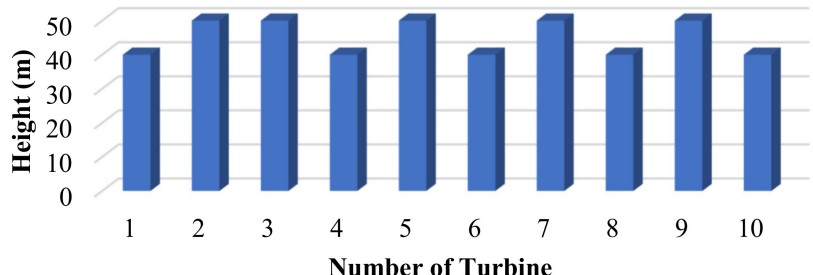

**Figure 8.** Heights of turbines in Case 1.

As shown in the figure, due to the low number of turbines and the abundant space between them, five turbines were placed with 40 m space between each and five other turbines were positioned with 50 m space. The objective function is 1.7214 megawatts per year and the total annual power generation is 5.5 megawatts.

*6.2. Second Case*

There are 15 turbines in this scenario and their positions and heights are displayed in Figures 9 and 10, respectively. In this case, Table 7 provides the coordination of turbines in Case 2.

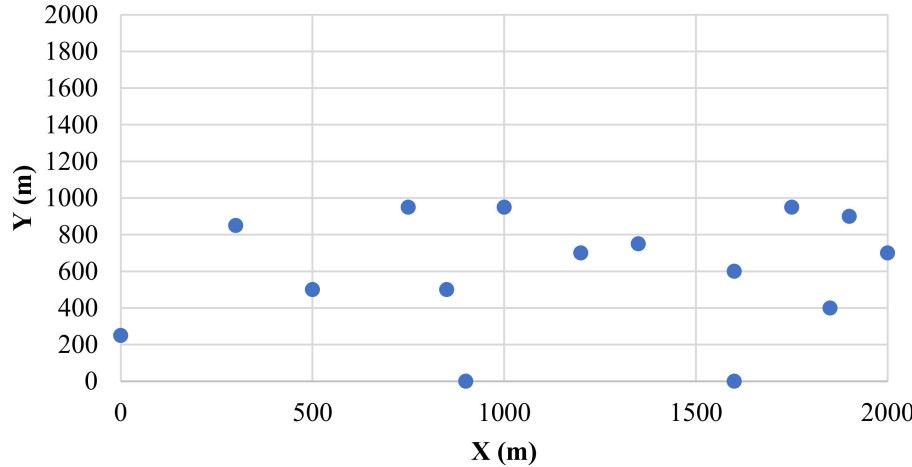

**Figure 9.** Position of turbines in Case 2.

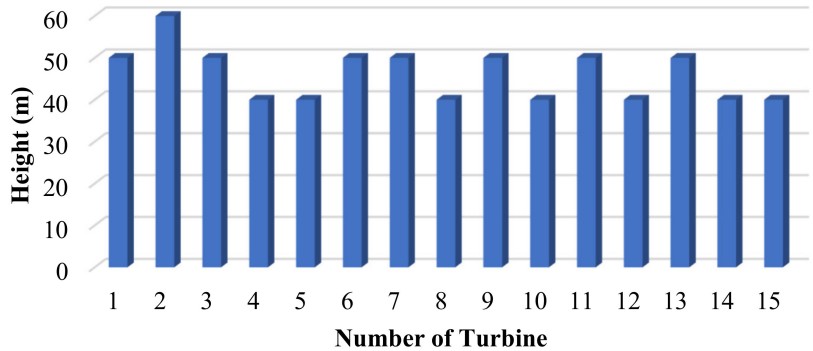

**Figure 10.** Height of each turbine in Case 2.

**Table 7.** Turbine coordinates in Case 2.

| Turbine Number | X (m) | Y (m) | Z (m) | Turbine Number | X (m) | Y (m) | Z (m) |
|---|---|---|---|---|---|---|---|
| 1 | 1850 | 400 | 50 | 9 | 1000 | 950 | 50 |
| 2 | 1600 | 0 | 60 | 10 | 1200 | 700 | 40 |
| 3 | 1900 | 900 | 50 | 11 | 700 | 2000 | 50 |
| 4 | 300 | 850 | 40 | 12 | 950 | 1750 | 40 |
| 5 | 750 | 950 | 40 | 13 | 1350 | 750 | 50 |
| 6 | 500 | 500 | 50 | 14 | 1600 | 600 | 50 |
| 7 | 850 | 500 | 50 | 15 | 0 | 250 | 40 |
| 8 | 900 | 0 | 40 | | | | |

The objective function is 1.784 per year per megawatt and the total power generation is 7.5 megawatts. There are eight turbines with a height of 50 m, 6 turbines with 40 m, and there is only one turbine with 60-m height. By comparing this case with Case Three, the power is decreased by 0.25 MW (equivalent to 3.22%), but the cost becomes 20% lower, which makes it cost-effective.

### 6.3. Case Three

In this case, there are 21 turbines in the Valfajr site in Manjil. The objective function is 2.2251 per year per megawatt. Because of a larger number of turbines compared to the previous cases, there are two turbines with a 60 m height, four with a 50 m height, and the rest have a height of 40 m. The total generated power is 7.75 megawatts. Figures 11 and 12 illustrate the placement and height of turbines in Case 3. In this case, Table 8 provides the coordination of turbines in Case 3.

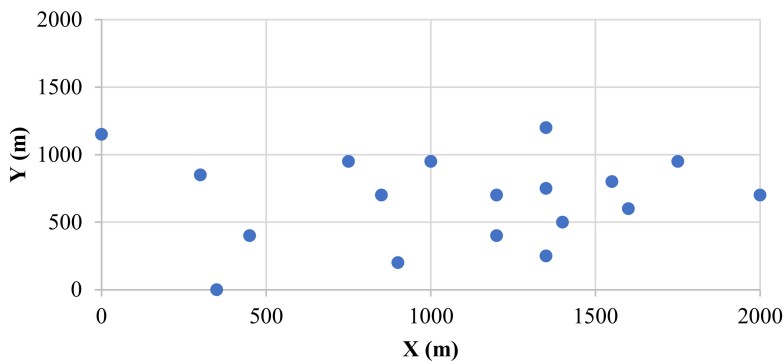

**Figure 11.** Position of turbines in Case 3.

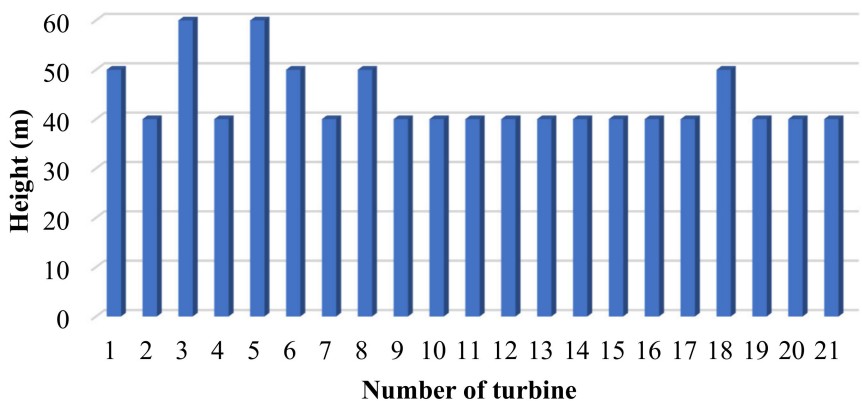

**Figure 12.** Height of turbines in Case 3.

**Table 8.** Turbine coordinates in Case 3.

| Turbines coordinate in Case 3 | | | | | | | |
|---|---|---|---|---|---|---|---|
| Number of Turbine | X(m) | Y (m) | Z (m) | Number of Turbines | X (m) | Y (m) | Z (m) |
| 1 | 1850 | 400 | 50 | 12 | 1750 | 950 | 40 |
| 2 | 1600 | 500 | 40 | 13 | 1350 | 750 | 40 |
| 3 | 1900 | 900 | 60 | 14 | 1600 | 600 | 40 |
| 4 | 300 | 850 | 40 | 15 | 1350 | 250 | 40 |
| 5 | 750 | 950 | 60 | 16 | 350 | 0 | 40 |
| 6 | 1400 | 500 | 50 | 17 | 450 | 400 | 40 |
| 7 | 850 | 700 | 40 | 18 | 0 | 1150 | 50 |
| 8 | 900 | 200 | 50 | 19 | 1350 | 1200 | 40 |
| 9 | 1000 | 950 | 40 | 20 | 1200 | 400 | 40 |
| 10 | 1200 | 700 | 40 | 21 | 1550 | 800 | 40 |
| 11 | 2000 | 700 | 40 | | | | |

Here, according to Bousejin et al. [76], as the layout of the turbines is not optimized, wind speed becomes reduced by 0.5 m/s due to the wake effect while the average speed is 14.5 m/s. In this case, the power generation is 6.77 megawatts and the objective function is 2.54 per year per megawatt. This means that optimization gives rise to a 10.75% increase in power generation and a 9.42% reduction in cost per unit of power. To compare power production at different speeds, the following diagram (Figure 13) is derived.

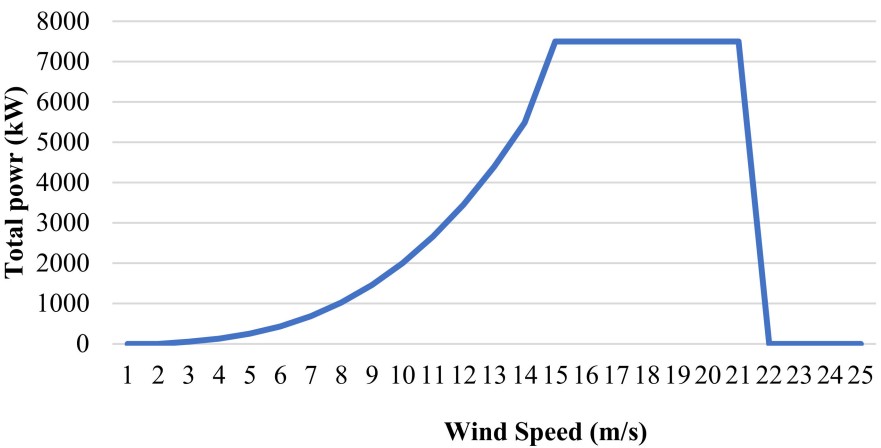

**Figure 13.** Total power for different wind speeds.

Moreover, the amount of energy unit cost is in correlation with wind speed. For the wind speed of 1 m/s, the unit cost of power becomes 320 per MW, which is quite high. As the wind speed increases to 10 m/s, the cost reduces drastically to 8.62 per MW, and as wind speed gains its rated value, which is 15 m/s, the unit cost reaches a minimum of 2.22 per MW (Figure 14). This implies that receiving high wind speeds are crucial for the wind farm to operate cost-effectively.

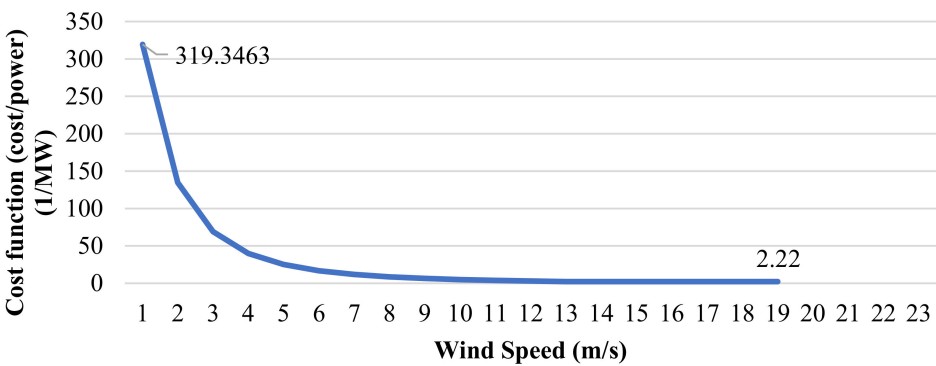

**Figure 14.** The unit cost of power with different wind speeds.

The 2-D Layout

In this section, the hub heights are considered constant and layout optimization for 21 turbines according to what was described in the previous section will be performed (Figure 15).

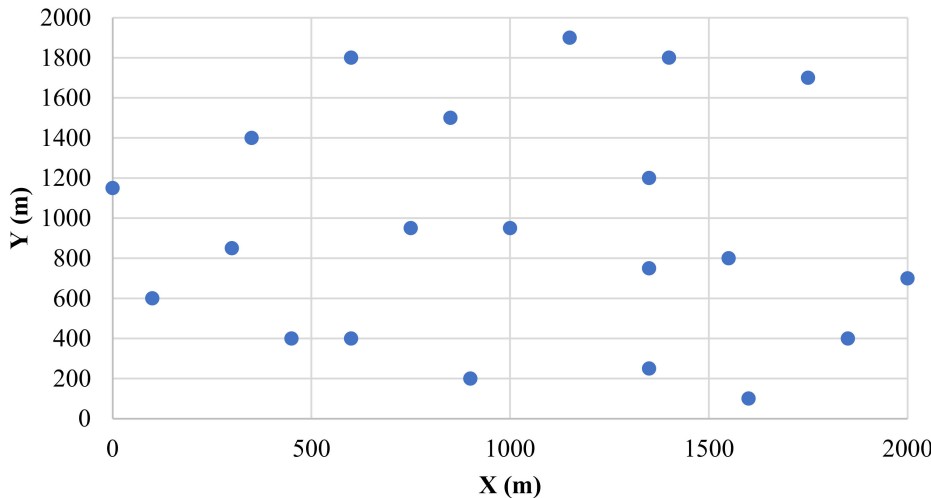

**Figure 15.** Layout of the 2-D case.

By comparing this layout with the previous one, it is obvious that there are $800 \times 2000$ m$^2$ of free space in 3-D optimization. The coordinates of the 2-D layout are given in Table 9.

By investigating and comparing the three scenarios, the following results achieved. The Mossetti's cost function is completely dependent on the number of turbines. It can be stated that by increasing the number of turbines the value of this function also increases, while the relationship between them is characteristically nonlinear. Furthermore, as the number of turbines grow, the function rises at a lower rate (Figure 16).

In comparison, the amount of linear growth between the increase in the number of turbines is determined. By increasing the number of turbines from 10 to 15, the wind turbine numbers are multiplied by 1.5, but the cost is multiplied by 1.335, which is less than a linear growth. Another discussion is about the objective function considered for this study. A general survey of the three scenarios is evaluated (Figure 17).

**Table 9.** Turbine coordinate in the 2-D case.

| Turbine Coordinates in the 2-D Case | | | | | |
|---|---|---|---|---|---|
| Number of Turbine | X (m) | Y (m) | Number of Turbine | X (m) | Y (m) |
| 1 | 1850 | 400 | 12 | 1750 | 1700 |
| 2 | 1600 | 100 | 13 | 1350 | 750 |
| 3 | 600 | 800 | 14 | 100 | 600 |
| 4 | 300 | 850 | 15 | 1350 | 250 |
| 5 | 750 | 950 | 16 | 350 | 1400 |
| 6 | 1400 | 1850 | 17 | 450 | 400 |
| 7 | 850 | 1500 | 18 | 0 | 1150 |
| 8 | 900 | 200 | 19 | 1350 | 1200 |
| 9 | 1000 | 950 | 20 | 600 | 400 |
| 10 | 1150 | 700 | 21 | 1550 | 800 |
| 11 | 2000 | 700 | | | |

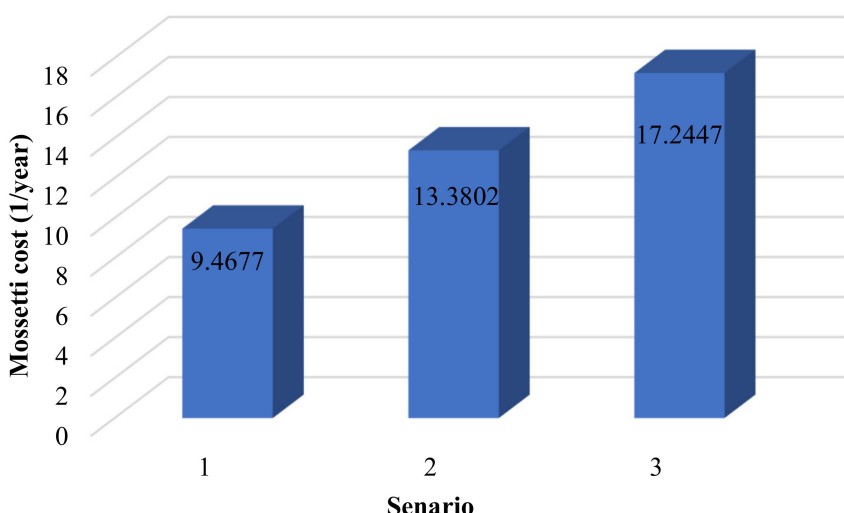

**Figure 16.** Cost of the wind farm in all cases.

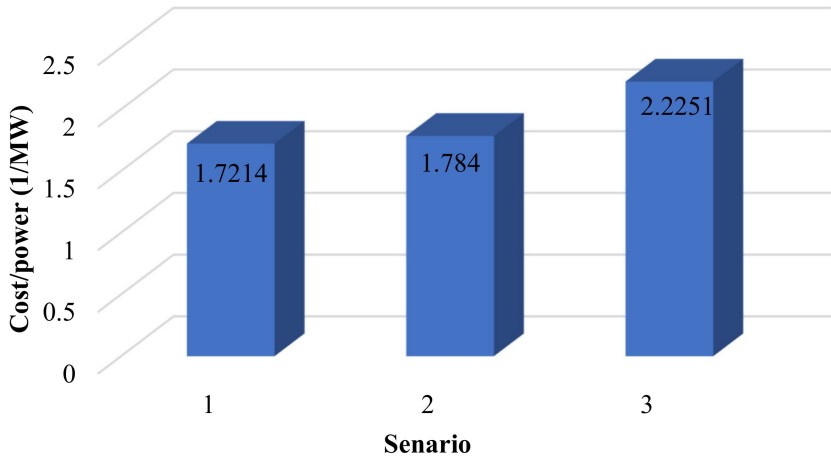

**Figure 17.** The objective function of three cases.

It is observed that the main objective function in this study decreases with the number of turbines. The reason is that the objective function is defined as the cost per power.

As the number of turbines increases, the generated power will increase. Considering that this number is in the denominator, as the number increases, the denominator will see a substantial growth, while the numerator, which is the Mossetti cost, is growing less steeply. Regarding power generation, by increasing the number of installed turbines, the total power will increase. This, however, creates a higher amount of wake effect, but when the layout optimized for zero wake effect, power generation would be even higher (Figure 18).

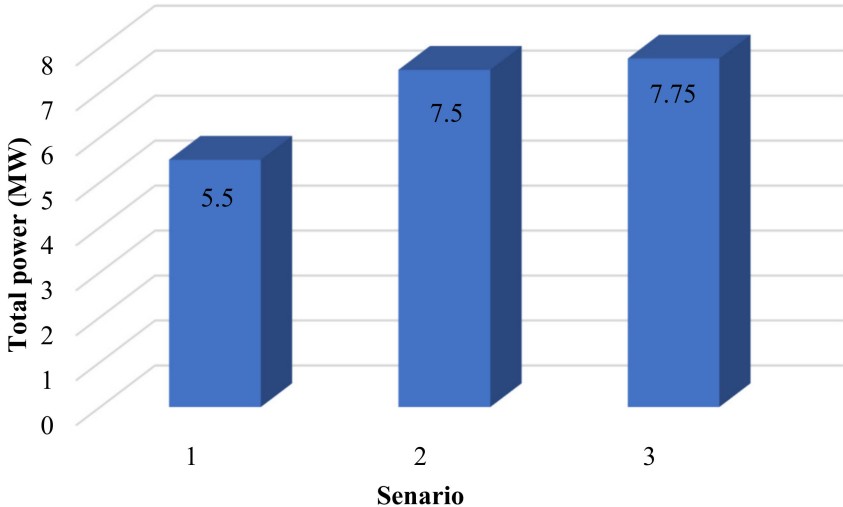

**Figure 18.** Power generation in all three cases.

## 7. Conclusions

The total power generated by wind farms is smaller than their theoretically calculated output. This is mainly because of the wake effect. In order to increase the actual output power of wind farms, the existing wake effect should be minimized. This puts a premium on the placement of wind turbines. In this study, the Valfarjr site of Manjil wind farm's layout is optimized in 3-D. In other words, wind turbine positions, wind farm configuration, and heights have been selected to be optimized in order to minimize the wake effect. Firstly, by considering the regime of wind and geographic data of the region, the wake effect is modeled by the Jensen's method. After that, the objective function, which is the cost to energy output, is calculated according to the method proposed by Mossetti. In the end, the objective function is optimized by employing the particle swarm optimization algorithm.

By setting optimal turbine coordinates, power generation increased by 10.75% and decreased the cost by 9.42%. Moreover, by using 15 turbines of 500 kW, installed in optimal positions and heights, power generation decreased by 3.22%, while the cost decreased by 20%.

For future studies, the economic aspect of the plant by using different economic methods could be investigated. Moreover, to achieve more accurate results, turbulent flow impacts can also be scrutinized.

**Author Contributions:** Conceptualization, methodology, software, validation, formal analysis, investigation, writing—original draft preparation, resources, data curation: M.Y., A.A., F.E., A.M. and S.H.; Writing—review and editing, visualization, supervision, project administration: R.D., S.H. and D.A.G. All authors have read and agreed to the published version of the manuscript.

**Funding:** This research received no external funding.

**Informed Consent Statement:** Not applicable.

**Data Availability Statement:** Provided data can be find by references or corresponding authors.

**Conflicts of Interest:** The authors declare no conflict of interest.

## Nomenclature

| **Acronyms** | | | |
|---|---|---|---|
| AD | Actuator Disc | $N_t$ | Number of Turbines |
| AEP | Annual Energy Production | R | Rotor's Radius (m) |
| BRCGA | Binary Real Coded Genetic Algorithm | V | Velocity (m/s) |
| CFD | Computational Fluid Dynamic | $V_{ij}$. | Fraction Perpendicular |
| COE | Cost of Energy | $P_n$ | Power (kW) |
| LS | Local Search | x | Distance (m) |
| PSO | Particle Swarm Optimization | y | Vector |
| PVS | Passing Vehicle Search | z | Hub Height (m) |
| WFLO | Wind Farm Layout Optimization | $z_0$ | Roughness Length (m) |
| **Symbols** | | | |
| A | Area (m$^2$) | **Greek symbols, Subscripts and Superscripts** | |
| c | Constant Learning Cfficient | $\theta$ | Angle (degree) |
| $C_p$ | Power Coefficient | $\rho$ | Air Density |
| $C_w$ | Wake Factor | . | Rate |
| k | Expansion Rate (-) | 0 | Ambient Conditions |
| m | Number of Turbines in a Row | shad | Covered Area |
| n | Number of Rows | | |

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
