# Peer review of "Wind Farm Layout Optimization with Different Hub Heights in Manjil Wind Farm Using Particle Swarm Optimization"

_applsci, doi:10.3390/app11209746_

Round 1

Reviewer 1 Report

In general, this paper shows the research gap and contributes to wind farm layout optimization, with distinct parameters and method. However, the authors should respond the following concerns:

Line 145. The quality of Figure 1 is not sufficient, particularly in writing Rw=Rr+K.x . In line 152, Formula (1) authors wrote k (in small letter), Please clarify.

Line 175. The quality of Figure 3 is not sufficient.

Line 245. Sounds this study not finished yet (will be investigated). Is this proposal or has been completed?

Line 254. The quality of Figure 5 is not sufficient.

Line 265. Figure 6 inconsistency in font size. Please modify. How to proceed if Cw>1?

Line 283. The interval and wind speed were chosen, but there is a lack of scientific reason to select this number. Any references or other considerations?

Line 301. Authors should add the unit for x-axis and y-axis.

Line 315. Similar case with Line 301

Line 361. Similar case with Line 301

Line 366. Updated with the Table 9.

Line 377. Please change senarios (x-axis) to Scenarios

Line 384. Similar case with Line 377

Line 395. Similar case with Line 377

Line 469-470. Inconsistency in writing the book title (all capital)

Line 487. Similar case with Line 469

Author Response

Answer Sheet

We would like to thank the Editor and the Reviewers for carefully examining our work and for providing us with the opportunity of revising and improving the manuscript. We have addressed all the comments and suggestions of yours and the reviewers and thoroughly modified the paper accordingly. Very comprehensive responses have been given to all reviewers, as you can see in this document. All the modifications are highlighted in the revised manuscript in order to facilitate the review process. Please see below our detailed response to every single comment raised. Thanking again for the attention given to this work, we look forward to hearing from you soon.

Sincerely yours,

The authors.

Reviewer #1:

  • Line 145. The quality of Figure 1 is not sufficient, particularly in writing Rw = Rr + K.x. In line 152, Formula (1) authors wrote k (in small letter), Please clarify.

Answer: We appreciate such a useful comment and we agree we the respected reviewer that the quality of figures ‎are ‎low. In this regard, we have enhanced the associated qualities (Please see figures).‎

  • Line 175. The quality of Figure 3 is not sufficient.

Answer: We appreciate such a useful comment and we agree we the respected reviewer that the quality of ‎figures ‎are ‎low. In this regard, we have enhanced the associated qualities (Please see figures).‎

  • Line 245. Sounds this study not finished yet (will be investigated). Is this proposal or has been completed?

Answer: Thanks for this valuable comment. We assure you that the proposal has been finished. This phrase was just typos and now corrected.

  • Line 254. The quality of Figure 5 is not sufficient.

Answer: We appreciate such a useful comment and we agree we the respected reviewer that the quality of ‎figures ‎are ‎low. In this regard, we have enhanced the associated qualities (Please see figures).‎

  • Line 265. Figure 6 inconsistency in font size. Please modify. How to proceed if Cw>1?

Answer: We appreciate such a useful comment and we agree we the respected reviewer that the quality ‎of ‎figures ‎are ‎low. In this regard, we have enhanced the associated qualities (Please see figures).‎ Also, for the montioned case (Cw>1), supplementary material are existed.

  • Line 283. The interval and wind speed were chosen, but there is a lack of scientific reason to select this number. Any references or other considerations?

Answer: Thanks for this valuable comment. The reasons to select this number are based on the literature and the availability of this data through the papers. It should be noted that, we have introduced scientific loops in the simulation codes to improve it in the optimization process.

  • Line 301. Authors should add the unit for x-axis and y-axis.

Answer: Thanks for this valuable comment. We have made the required modifications.‎

  • Line 315. Similar case with Line 301

Answer: Thanks for this valuable comment. We have made the required modifications.

  • Line 361. Similar case with Line 301

Answer: Thanks for this valuable comment. We have made the required modifications.‎

  • Line 366. Updated with the Table 9.

Answer: Thanks for this valuable comment. We have made the required modifications.‎

  • Line 377. Please change scenarios (x-axis) to Scenarios

Answer: Thanks for this valuable comment. We have made the required modifications.‎

  • Line 384. Similar case with Line 377

Answer: Thanks for this valuable comment. We have made the required modifications.‎

  • Line 395. Similar case with Line 377

Answer: Thanks for this valuable comment. We have made the required modifications.‎

  • Line 469-470. Inconsistency in writing the book title (all capital)

Answer: Thanks for this valuable comment. We have made the required modifications.‎

  • Line 487. Similar case with Line 469

Answer: Thanks for this valuable comment. We have made the required modifications.‎

Reviewer 2 Report

The manuscript considers wind regime and geographic data of a case study, and using Jensen's method, the wake effect of the turbines’ configuration further is simulated. Authors propose objectives function in the optimization problem to be set to find the optimal coordinate of the wind turbines and the cost of electricity generation based on Mossetti cost function by application through particle swarm optimization method. Several versions would be essential to improve the quality of the manuscript.

Nevertheless, the topic of the paper is attractive, and some of the results are remarkable. Of course, the following items are recommended to improve the research. 

+ The manuscript should include "materials and methods" section. Section 2 must be devoted to "materials and methods", where all the methods and data is described. Mathematical model must be a subsection of materials and methods. 

+ figure 1 must be reproduced and the caption should not include citation.

+ the equations to wake shadow area and further equations must include citation.

+ Native poofreading to check the English is required. The paper suffers from several typos and mistakes. I suggest that one author should go through the whole paper to make sure of the flow.

+Provide units for presented values through the manuscript.

+Also upgrade the introduction section with latest literature.

+ After presenting each equation, it is necessary to provide the references used and define the presented parameters

+ Some sentences have been repeated throughout the article. Please correct them.

+ elaborate on the validation

+ in figure 14 spelling of the "speed" is not correct.

+How do you explain the relation between cost function and wind speed in figure 14?

+ elaborate on the future research, how the results can be improved?

Author Response

Answer Sheet

We would like to thank the Editor and the Reviewers for carefully examining our work and for providing us with the opportunity of revising and improving the manuscript. We have addressed all the comments and suggestions of yours and the reviewers and thoroughly modified the paper accordingly. Very comprehensive responses have been given to all reviewers, as you can see in this document. All the modifications are highlighted in the revised manuscript in order to facilitate the review process. Please see below our detailed response to every single comment raised. Thanking again for the attention given to this work, we look forward to hearing from you soon.

Sincerely yours,

The authors.

Reviewer #2

  • The manuscript should include "materials and methods" section. Section 2 must be devoted to "materials and methods", where all the methods and data is described. Mathematical model must be a subsection of materials and methods.

Answer: The title of the second section changed to "materials and methods", since, as this study is a numerical one, all the methods and data are presented within this section.

  • Figure 1 must be reproduced and the caption should not include citation.

Answer: Thanks for this valuable comment. We have made the required modifications.‎

  • The equations to area and further equations must include citation.

Answer: Thanks for this valuable comment. We have made the required modifications.

  • Native proofreading to check the English is required. The paper suffers from several typos and mistakes. I suggest that one author should go through the whole paper to make sure of the flow.

Answer: Thanks for this valuable comment. We have made the required modifications.

  • Provide units for presented values through the manuscript.

Answer: Thanks for this valuable comment. We have made the required modifications.‎

  • Also upgrade the introduction section with latest literature.

Answer. Dear respected reviewer, thanks for your valuable comment. We have improved the quality of the introduction ‎section as listed below. We hope such comprehensive ‎modifications could satisfy the reviewer’s concerns.‎

  • The revised introduction is much more coherent.‎
  • The used references are updated.‎
  • We have added more discussion on the feasibility and challenges of similar works.‎
  • We improved the structure of the introduction section.‎
  • After presenting each equation, it is necessary to provide the references used and define the presented parameters.

Answer. Dear respected reviewer, thanks for your valuable comment. In this regard, we have made the essential corrections.

  • Some sentences have been repeated throughout the article. Please correct them.

Answer. Thank you for your warm words about our work and this useful comment. We completely revised the ‎English language of the manuscript and we modified all typos. The English language is currently acceptable ‎and we hope it could satisfy the reviewer’s concern.‎

  • Elaborate on the validation

Answer. Thanks for this valuable comment. More explanations are added.

  • In figure 14 spelling of the "speed" is not correct.

Answer. Thanks for this valuable comment. We have made the required modifications.

  • How do you explain the relation between cost function and wind speed in figure 14?

Answer. More explanations are added to clarify their relationship. This part is highlighted by blue.

  • Elaborate on the future research, how the results can be improved?

Answer: More explanations are added. This part is highlighted by blue.

Round 2

Reviewer 2 Report

All comments had been addressed.